# Molecular Epidemiology of Carbapenem-Resistant *Klebsiella aerogenes* in Japan

**DOI:** 10.3390/ijms25084494

**Published:** 2024-04-19

**Authors:** Kentarou Takei, Miho Ogawa, Ryuji Sakata, Hajime Kanamori

**Affiliations:** 1Department of Infectious Diseases, Internal Medicine, Tohoku University Graduate School of Medicine, Sendai 980-8575, Japan; kentarou.takei.e7@tohoku.ac.jp; 2Department of Bacteriology, BML Inc., Kawagoe 350-1101, Japan

**Keywords:** *Klebsiella aerogenes*, carbapenemase producing, carbapenemase resistant, whole-genome sequencing, plasmid, integron, transposon

## Abstract

Information regarding *Klebsiella aerogenes* haboring carbapenemase in Japan is limited. A comprehensive nationwide survey was conducted from September 2014 to December 2022, and 67 non-duplicate strains of carbapenem-resistant *K. aerogenes* were isolated from 57 healthcare facilities in Japan. Through genetic testing and whole-genome sequencing, six strains were found to possess carbapenemases, including imipenemase (IMP)-1, IMP-6, New Delhi metallo-β-lactamase (NDM)-1, and NDM-5. The strain harboring *bla*_NDM-5_ was the novel strain ST709, which belongs to the clonal complex of the predominant ST4 in China. The novel integron containing *bla*_IMP-1_ featured the oxacillinase-101 gene, which is a previously unreported structure, with an IncN_4_ plasmid type. However, integrons found in the strains possessing *bla*_IMP-6_, which were the most commonly identified, matched those reported domestically in *Klebsiella pneumoniae*, suggesting the prevalence of identical integrons. Transposons containing *bla*_NDM_ are similar or identical to the transposon structure of *K. aerogenes* harboring *bla*_NDM-5_ previously reported in Japan, suggesting that the same type of transposon could have been transmitted to *K. aerogenes* in Japan. This investigation analyzed mobile genetic elements, such as integrons and transposons, to understand the spread of carbapenemases, highlighting the growing challenge of carbapenem-resistant *Enterobacterales* in Japan and underscoring the critical need for ongoing surveillance to control these pathogens.

## 1. Introduction

*Klebsiella aerogenes* is a Gram-negative, rod-shaped, facultative bacteria belonging to the *Enterobacterales*. *K. aerogenes* was previously classified under the *Enterobacter* genus alongside the *Enterobacter cloacae complex*, which has emerged as a nosocomial pathogen [1]. Whole-genome sequencing (WGS) of multidrug-resistant isolates led to the reclassification of a species into the genus *Klebsiella*, currently named *K. aerogenes* [2].

This bacterium, which basically carries the AmpC β-lactamase on its chromosome, was once a main pathogen in nosocomial infections [3]. However, a decreasing trend in its prevalence has been reported since 2010 in Western European countries, such as France. This is due to the dramatic rise in the *Escherichia coli* pandemic clone O25:H4-ST131, along with *Klebsiella pneumoniae* and *Enterobacter cloacae complex*, extended-spectrum β-lactamase (EBSL), and/or carbapenemase-producing strains [4]. The major *Enterobacterales* that also cause nosocomial infections in China and the United States are *E. coli*, *K. pneumonia*, and *E. cloacae complex*; *K. aerogenes* is less frequent [5,6].

Gram-negative bacteria, which are the causative agents of nosocomial infections, exhibit resistance to multiple drugs and have a built-in ability to find new ways to acquire resistance [7]. Currently, one of the most concerning drug-resistant organisms is carbapenem-resistant *Enterobacterales* (CRE), a notable contributing factor of which is the presence of carbapenemase, which propagates through conjugative transfer via plasmids [8]. Globally, carbapenemase-producing *K. pneumoniae* and *E. cloacae complex* are the central focus of discussions on CRE [9].

Notwithstanding global trends, in Japan, the scenario is somewhat different. Interestingly, non-carbapenemase-producing *K. aerogenes*, which is the causative bacterium of CRE infections, continues to be the predominant pathogen in Japan [10]. Epidemiological studies conducted in major Japanese hospitals support this trend [11]. From another point of view, this indicates that carbapenemase-producing *K. aerogenes* strains are extremely rare or non-existent. Specifically, the National Institute of Infectious Diseases (NIID) reported a total of 1684 cases of CRE infections in 2018, of which 631 (37.5%) were caused by *K. aerogenes*. However, among the carbapenem-resistant *K. aerogenes* strains, only two strains (0.32%) were identified as having carbapenemases; specifically, the imipenemase (IMP) type, which is the most common, is considered to be endemic to Japan [10,12]. Although certainly rare in frequency, *K. aerogenes* has always been associated with poor outcomes in invasive infections, particularly bloodstream infections, and the presence of carbapenemases in this bacterium demands even greater caution [13].

With the attention focused on carbapenemase-producing *K. aerogenes*, it is notable that *K. aerogenes* harboring New Delhi metallo-β-lactamase (NDM)-5 was recently reported for the first time in patients from remote areas within the country without any history of overseas travel [14]. This suggests the potential presence of *K. aerogenes* strains carrying not only the prevalent IMP-type carbapenemase but also an internationally derived NDM type of carbapenemase potentially lurking in the community or community hospitals [15].

In Japan, the definition of CRE within the country was established in September 2014 under the Infectious Diseases Control Law. Since then, many surveys in the country, including our previously conducted CRE nationwide survey, have been conducted primarily based on that definition [16].

In this study, we performed molecular epidemiological analysis of carbapenemase-producing *K. aerogenes* strains collected from medical institutions nationwide via a testing center, with the aim to elucidate the type and its transmission of carbapenemase genes and clarify the spread of major carbapenemases by analyzing mobile genetic elements, such as integrons and transposons.

## 2. Results

### 2.1. Carbapenem-Resistant Klebsiella aerogenes Isolates across Medical Facilities in Japan

A total of 67 non-duplicate clinical isolates of carbapenem-resistant *K. aerogenes* from 67 individuals, ranging in age from 0 to 97 years, were collected from 57 medical institutions in 20 prefectures in Japan between September 2014 and December 2022. The Kanto region, including Tokyo, had the largest number of strains, accounting for 59.7% (40/67) of the total, and overall, many of the strains were from eastern Japan (Figure 1). The mean age of the patients from whom each specimen was derived was 75.8 years [95% confidence interval (CI): 71.8–79.9], with a median age of 79 years, including 44 men and 23 women. Among the surveyed medical institutions, the average bed capacity, excluding outpatient clinics, was 198 beds [95% CI: 157–238], with a median bed count of 160. In categorizing the types of healthcare facilities, there were 2 outpatient-only clinics, 41 hospitals that predominantly focused on acute care, 4 dedicated to rehabilitative care, and 10 that catered to long-term or chronic care. Basic information and genetic test results regarding various *K. aerogenes* strains harboring carbapenemase are demonstrated in Table 1.

### 2.2. PCR, Whole-Genome Analysis, and MLST

Among the 67 carbapenem-resistant isolates, six tested positive for carbapenemase genes by Polymerase Chain Reaction (PCR), with four testing positive for the IMP gene (*bla*_IMP-1 group_) and two for the NDM gene (*bla*_NDM group_). Over approximately 8 years, the proportion of carbapenemase-producing *K. aerogenes* among CRE was 9.0% (6/67). WGS was performed on the six PCR-positive strains, revealing that the carbapenemase-producing genes were IMP-1, IMP-6, NDM-1, and NDM-5 (*bla*_IMP-1_, *bla*_IMP-6_, *bla*_NDM-1_, and *bla*_NDM-5_, respectively) using bioinformatic tools. The profiles of the six strains, including those of the other antibiotic resistance genes, are summarized in Table 1.

Referring to the results of drug susceptibility, all strains except TUT0037 were placed on CRE due to resistance to meropenem; TUT0037 was not tested for meropenem susceptibility but was placed on CRE because of drug resistance to imipenem/cilastatin and cefmetazole. Although the strains were observed to harbor the quinolone resistance genes *oqxA*, *dfrA14*, and *qnrD1*, all but TUT0015 showed susceptibility to levofloxacin. TUT0013, TUT0023, and TUT0032 harbored the sul1 gene, but only TUT0023 was resistant to the drug. The results of the drug susceptibility testing for the six strains are listed in the Appendix A (Appendix A).

### 2.3. MLST and Phylogenetic Analysis of Carbapenemase-Producing Klebsiella aerogenes

The six *K. aerogenes* isolates were classified into five STs using multilocus sequence typing (MLST). These belonged to STs 4, 189, 296, 668, and 709. ST709 was a novel detection (Table 1). Two ST668 strains were isolated from different patients in the same hospital, albeit in different years. According to BURST analysis based on allelic profiles, ST4, ST296, ST668, and ST709 were included in three clonal complexes (CCs). ST189 did not constitute a CC. ST4 and ST668 were identified as potential ancestral types (AT); ST296 was included as part of a CC with ST434 as an AT (Figure 2). Overall, there was no uniformity in the STs, and their origins were diverse.

### 2.4. Plasmid Type (Inc Genes)

The TUT0015 strain was found to have *IncN_4_* on the same contig containing *bla*_IMP-1_. Additionally, two strains harboring *bla*_NDM_ also harbored IncX_3_. The two strains TUT0023 and TUT0032, which were isolated in Osaka, were discovered to harbor the IncN and IncR genes only through WGS. However, no Inc gene was identified in the contig containing *bla*_IMP-6_ for these strains. Similarly, no *Inc* gene was identified in the contig containing *bla*_IMP-6_ for the strain isolated in Fukuoka (TUT0013), even though it had the IncFIA, IncFII, IncH1b, and IncN genes according to WGS.

### 2.5. Integron Structure Harboring bla_IMP_

The integron structure was identified in the four strains carrying the *bla*_IMP_ gene. Of these*,* three strains had a complete integron structure, while one had a structure known as clusters of attC sites lacking integron integrases (CALIN), which are characterized by the absence of a class 1 integrase gene (*intI1*) (Figure 3).

Regarding the three strains harboring *bla*_IMP-6_, the integron structure of the isolate in Fukuoka in 2016 (TUT0013) consisted of *intI1* at the forefront, which was followed downstream by *acc(6*′*)-Ib*, *bla*_IMP-6_, *aadA2*, Quaternary ammonium compound resistance protein E delta 1(*QacEΔ1*), and *sul1* with three attC sites (71 bp, 126 bp, and 59 bp). This structure was identified by matching the integron (In1690). The integron of TUT0032 was identical to the integron structure (In1547). Though TUT0023 was missing *intI1*, the gene cassettes for integrons containing *bla*_IMP-6_ were the same for TUT0023 and TUT0032. Unlike the aforementioned three integrons containing *bla*_IMP-6_, the integron structure containing *bla*_IMP-1_ of the isolate in Fukuoka in 2016 was different. This integron had *aph(3*′*)-XV*, *bla*_OXA-101_, an efflux small multidrug resistance (SMR) transporter gene, *bla*_IMP-1_, and *aadA5* with four attC sites (107 bp, 63 bp, 98 bp, and 126 bp) downstream of *intl1*. Although this integron structure was searched for using Integrall and the NCBI Basic Local Alignment Search Tool (BLAST) for plasmids with a similar cassette structure, no identical structure was identified (Figure 3).

### 2.6. Transposons Containing bla_NDM-1_ and bla_NDM-5_ on IncX_3_-Type Plasmid

Two strains of *K. aerogenes* harboring *bla*_NDM_ were identified in different regions. Strains with *bla*_NDM-5_ were isolated in Ishikawa in 2018, and those with *bla*_NDM-1_ were isolated in Kanagawa in 2019. Referring to the gene sequence, Tn*2*, IS*3000*, and ISA*ba125* were commonly located upstream of both *bla*_NDM-1_ and *bla*_NDM-5_, similar to TUT0037 and TUT0049. ISA*ba125* and IS*3000*, constituting Tn*125* and Tn*3000*, respectively, were transposases involved in transposon transposition events (Figure 4). However, a paired identical transposase located downstream of the *bla*_NDM_ was not identified; a composite transposon (CT) could not be confirmed. Genetic abbreviations related to mobile genetic elements are summarized in the Appendix A.

## 3. Discussion

In this study, (1) we conducted a nationwide survey of carbapenem-resistant *Klebsiella aerogenes* in Japan to determine the frequency of carbapenemase-producing strains, as well as the facilities and regional characteristics from which they were isolated; (2) through whole-genome analysis, we revealed information on prevalent carbapenemase types and STs of *K. aerogenes* across Japan; and (3) we identified the integrons and transposons encoding carbapenemases and investigated their origins.

### 3.1. Global and Local Perspectives on Carbapenemase Presence in Klebsiella aerogenes

Information on carbapenemase possession among *K. aerogenes* strains worldwide is limited. First of all, *K. aerogenes* has been reported to possess essentially small amounts of carbapenemase. Chromosomal AmpC overproduction and inherent properties have been suggested as possible reasons for the low number of carbapenemase-carrying strains [17]. In a survey of 151 institutions in 35 countries, the rate of carbapenemase possession among multidrug-resistant *K. aerogenes* strains was reported to be 3.1% [18]. In the two previous reports from China, the positive proportion for carbapenemase ranged from 8.3% to 14.3% among CRE [19,20]. In Japan, the recent NIID annual report shows that carbapenemase-producing *K. aerogenes* was not reported in some years, and the frequency of its official annual reporting as a cause of infection was estimated to be about 0–0.32% each year [12,21,22,23]. In the current study, the proportion of carbapenemase-producing bacteria among CRE in *K. aerogenes* was higher than officially reported, accounting for 9%. This may have been due in part to the inclusion of colonization, as well as the bacteria that necessarily caused the infections in this study. Information about the target medical facility is also important and may have contributed to the high carbapenemase proportion. In Japan, 69% of medical institutions with inpatient facilities are small, with less than 200 beds, and these small facilities are often left out of public surveys [24]. The full picture of bacterial resistance in these small healthcare facilities is not known. In this study, the average number of beds in 55 facilities, excluding clinics, was 198, with 63.4% (35/55) having fewer than 200 beds. This proportion is similar to the proportion of hospitals classified by number of beds in Japan described above, suggesting that it could reflect the actual situation of small hospitals in Japan.

### 3.2. Sequence Types and Carbapenemase Variants in K. aerogenes

Five STs were identified in this study: STs 4, 189, 296, 668, and 709. ST4 is one of the predominant STs in China and has been reported in nosocomial clusters in pediatric hospitals and among men who have sex with men [25,26]. While the international travel history of the patient from whom it was isolated in our study is unknown, the novel detection of ST709, which is the CC of ST4, raises the question of whether this strain will be reported internationally or domestically in the future. However, further studies are needed to confirm this hypothesis. ST189 has been registered in Brazil, but details regarding it are unclear. ST296 was registered in China but was discovered in Brazil before 2016 [27]. Two strains isolated from Osaka belonged to ST668 reported domestically and originated from the same hospital. ST668 has been previously registered in Japan, and given its central position within the CC, it is plausible that ST668 potentially represents one of the prevalent STs in the country. Because of the lack of comprehensive studies on *K. aerogenes* in Japan, including genetic testing, the mainstream STs are unknown, but some STs of the strains carrying carbapenemase could be clarified. In addition to STs, the type of carbapenemase could be mentioned. The most common carbapenemase gene identified in this study was *bla*_IMP-6_, followed by *bla*_IMP-1_, *bla*_NDM-1_, and *bla*_NDM-5_ in order of frequency. According to official reports in 2018, the most prevalent carbapenemase is IMP (85.5%), followed by NDM (10.4%) and KPC (3.4%) [10,12]. The IMP type, which is the most common, is considered to be endemic to Japan. On the other hand, the NDM type was first reported in 2011 and the KPC type in 2014, both of which were classified as carbapenemases of foreign origin [28,29]. Although in this study, the total number of isolates harboring carbapenemase was low, it was noteworthy that relatively more isolates, i.e., 2/6 (33.3%), harbored *bla*_NDM_. Owing to the small number of strains, no significant trend was observed in typing with MLST. In addition, no clear trend in the association between the STs and carbapenemase was evident.

### 3.3. Mobile Genetic Elements in This Study, Including Plasmids, Integrons, and Transposons

Plasmids carrying these carbapenemases and information on integrons and transposons provide valuable insights into the origins of carbapenemase genes [30,31].

As integrons related to *bla*_IMPs_ in *Enterobacterales*, two *bla*_IMP-1_-containing integrons (In1311 and In1312) have been internationally reported [32]. From China, integron (In1223) has been reported [33]. Moreover, another integron from *Enterobacter hormaechei* (In1426) and an integron from *E. coli* (In798) have been reported from Japan [34,35] (Appendix A). The structures of these integrons harboring *bla*_IMP-1_ are more complex and diverse, varying both nationally and internationally, as well as between different bacterial species, compared with integrons harboring *bla*_IMP-6_ (In1321, In1547, and In1690), which are less frequently reported outside of Japan [32,36]. In this study, the integron harboring *bla*_IMP-1_ with *bla*_OXA-101_ inserted could not be found by BLAST or Integrall. In addition to the integron, TUT0015 showed interesting Inc-type characteristics. IncN_4_ was previously reported in Italy as a plasmid of *Citrobacter freundii* carrying *bla*_OXA-181_ (GenBank accession number JQ996149). We did not find any reports on the association between the IncN_4_ plasmid type and *bla*_IMP-1_. There is no information available on the plasmids of this Inc type or the integrons they carry, and further research is needed in this area.

As for IMP-6, the epidemic is mostly limited to western Japan [37]. Regarding our three strains harboring *bla*_IMP-6_, the integron of the strain isolated in Fukuoka in 2016 (TUT0013) was In1690, which is one of the representative integrons harboring *bla*_IMP-6_ and reported in several studies [35,36,38]. The integron (In1547) of TUT0032 was similar to that of In1690, although a transposase (ISV*sa10*) insertion was observed downstream of aadA2. TUT0023 also lacked the class 1 integrase of In1547, but otherwise, the integron structure was identical. These results suggest that the integrons harboring *bla*_IMP-6_ identified in this study are of a type that is relatively common in Japan and could have been transmitted by *E. coli* or *K. pneumoniae*, which are more common hosts of *bla*_IMP-6_ [39].

In addition to *bla*_IMP_ carbapenemases, we isolated two bacterial strains that produce *bla*_NDM_ carbapenemases. For *bla*_NDM_, unlike *bla*_IMP,_ the involvement of transposons in the mobile elements of the resistance genes is important [40]. Recently, two strains of *K. aerogenes* harboring *bla*_NDM-5_ were reported for the first time in Japan [14]. These strains were isolated between 2019 and 2020, which coincided with or slightly followed the period of our isolates. According to the previously mentioned report, when the gene structure was related to *bla*_NDM-5_, the 5′ end of the *bla*_NDM-5_ sequence in the *K. aerogenes* NDM-5 plasmid consisted of IS*5*, ISA*ba125*, IS*3000*, and Tn*2*. This structure was very similar to our two *bla*_NDM_ *K. aerogenes* strains (TUT0037, TUT0049), although IS*5* was missing in the plasmid carrying *bla*_NDM-1_ in our case. The plasmids reported in China harboring *bla*_NDM-5_ (Genbank accession no. MK450346) and those reported in Japan harboring *bla*_NDM-5_ (Genbank accession no. LC54851 and AP019679) also had the same IS element and/or transposase, such as IS*5*, ISA*ba125*, IS*3000*, and Tn*2* at the 5′ end of *bla*_NDML-5_, and both were on IncX_3_-type plasmids. The structure containing “Tn*2-*IS*3000*-ISA*ba125-*IS*5*” at the 5′ end of *bla*_NDM_ was suggested to be possibly related to IncX_3_. Moreover, *IS3000* and *ISAba125* are considered components of transposons, such as Tn*3000* and Tn*125*. Tn*3000* is typically a complex transposon consisting of two copies of IS*3000* and is known to contain the structure of Tn*125* [41,42]. Although the composite transposon (CT) structure could not be confirmed, the *bla*_NDM_-containing structures of TUT0034 and TU0049 suggest that they could be part of the Tn*3000* transposon. Tn*3000* is common in the genus *Klebsiella* and is often linked to South Asian origins, especially India, more so than East Asia, with frequently undefined replicon types [40].

This study had several limitations. Owing to the limitations of short-read sequencing, we were unable to elucidate the entire structure of the plasmids, let alone the mobile genetic elements. A more detailed investigation of these structures would require long-read sequencing. Nevertheless, we believe that we were able to discuss the origins of integrons and transposons based on the limited information available. Second, because preserved strains that meet the national criteria for CRE were obtained from among the strains requested for microbiological testing by each medical institution through major testing companies in Japan, detailed patient information (such as medical history, travel history, and outcomes) was not available, and we could not determine whether the isolated strains were the causative agents of infection or merely colonizers.

## 4. Materials and Methods

### 4.1. Bacterial Strains

Nonduplicate clinical isolates of carbapenem-resistant *K. aerogenes* were collected from domestic medical institutions in Japan between September 2014 and December 2022. To avoid the duplication of samples from any single individual, only the first isolate detected in each patient was selected. The identification of bacterial strains and antimicrobial susceptibility test were conducted using Vitek MS (BioMérieux, Marcy l’Étoile, France) and the Microscan WalkAway^®^ system (Beckman Coulter, Brea, CA, USA), with accompanying panels of the Microscan Neg^®^ series (Neg Combo EN 4 J and Neg MIC EN 2 J) (Beckman Coulter, Brea, CA, USA). The bacterial solution was prepared using a prompt inoculation method [43]. Antimicrobial susceptibility was tested according to the guidelines outlined by the Clinical and Laboratory Standards Institute (CLSI) [44,45,46,47]. CRE was defined in accordance with the Japanese Infectious Disease Control Law as *Enterobacterales* isolates exhibiting a minimum inhibitory concentration (MIC) of ≥2 µg/mL for meropenem, ≥2 µg/mL for imipenem, and ≥64 µg/mL for cefmetazole. Information regarding the size and role of the targeted medical institutions was compiled based on a survey of each hospital’s publicly available hospital profile, number of beds, and bed classification (Appendix A).

### 4.2. Screening of Carbapenemase Genes by PCR

Carbapenemase genes were screened by using the Cica Geneus^®^ Carbapenemase Genotype Detection KIT2 (Kanto Chemical Co., Tokyo, Japan), which can detect the following genes: *bla*_IMP-1,_ Verona integron-encoded metallo-beta-lactamase (*bla*_VIM_), Guiana extended-spectrum (*bla*_GES_), *bla*_KPC_, *bla*_NDM_, *bla*_OXA-48_, and *bla*_IMP-6_. Bacterial processing and thermal cycling were performed according to the manufacturer’s instructions.

### 4.3. Whole-Genome Sequencing and Genomic Analysis

DNA was extracted using the QIAamp DNA Mini Kit (QIAGEN, Hilden, Germany), and the library was prepared using the Nextera DNA Flex Library Prep Kit (Illumina, San Diego, CA, USA), employing the Nextera DNA CD Index as the index adapter, as per the manufacturer’s guidelines. Genomic sequencing of *K. aerogenes* strains positive for carbapenemase genes by PCR was performed on an Illumina iSeq 100 (Illumina, San Diego, CA, USA), utilizing paired-end 150 bp reads.

The sequencing data obtained were annotated using the DNA Data Bank of Japan (DDBJ) Fast Annotation and Submission Tool (DFAST) [48]. Acquired antibiotic resistance genes were identified using ResFinder 4.4.2 (http://genepi.food.dtu.dk/resfinder [accessed on 10 November 2023]) from the Center for Genomic Epidemiology (CGE) [49,50]. Because the *bla*_IMP-1_ and the *bla*_IMP-6_ differ by only one base (base 640 of 741 bp in length: adenine for *bla*_IMP-1_ and guanine for *bla*_IMP-6_), the detailed identification of *bla*_IMP-1_ and *bla*_IMP-6_ was confirmed using the NCBI BLAST database (https://blast.ncbi.nlm.nih.gov/Blast.cgi [accessed on 10 November 2023]) using the nucleotide sequences of their respective representative reference isolates (GenBank accession nos. NG_049172.1 and NG_049220.1), which ultimately led to their identification.

### 4.4. MLST and Phylogenetic Analysis of Carbapenemase-Producing K. aerogenes

The carbapenemase-producing *K. aerogenes* isolates were analyzed by the amplification of seven housekeeping genes (dnaA, fusA, gyrB, leuS, pryG, rplB, and rpoB). STs were assigned by querying the Institut Pasteur’s *Klebsiella aerogenes* PubMLST database (https://pubmlst.org/organisms/klebsiella-aerogenes [accessed on 15 November 2023]). CCs were defined at the single-locus variant level. STs and CCs were assessed using BURST analysis via a PubMLST plugin.

### 4.5. Analysis of Plasmids, Integrons, and Transposons Associated with Carbapenemase

Inc genes in the plasmids were identified using PlasmidFinder 2.1 (https://cge.food.dtu.dk/services/PlasmidFinder/ [accessed on 10 November 2023]) from the CGE [50,51]. To identify plasmids related to carbapenemase, contigs containing carbapenemase were selectively chosen for analysis. Integron and transposon regions were confirmed using Integron Finder 2.0 and VRprofile2 (https://tool2-mml.sjtu.edu.cn/VRprofile/ [accessed on 7 January 2024]) [52,53]. Specific configurations and sequences of integrons were identified using Integrall (http://integrall.bio.ua.pt/ [accessed on 12 December 2023]) [54]; for configurations and sequences not listed in Integrall, NCBI BLAST was used to search for sequences containing structures around carbapenemase. ISfinder (https://www-is.biotoul.fr/index.php [accessed on 12 December 2023]) and TnCentral (https://tncentral.ncc.unesp.br/ [accessed on 25 January 2024]) were used supplementally to confirm IS elements and transposase, respectively [55,56].

### 4.6. Data Availability

The WGS data were deposited in GenBank via the DDBJ. BioProject accession number: PRJDB17235. The Biosample numbers for the six strains were as follows: SAMD00670838, SAMD00728333, SAMD00728334, SAMD00728335, SAMD00728336, and SAMD00728337.

### 4.7. Statistical Analysis

We performed a chi-squared test to check the proportion of the number of carbapenem-resistant *K. aereogenes* strains in relation to the main function of the hospital and the number of beds. Analysis was performed using JMP Pro 16 statistical analysis software (SAS Institute, 2021, Cary, NC, USA). Differences were considered significant at a corrected *p*-value < 0.05.

## 5. Conclusions

Over an 8-year period, we identified 67 carbapenem-resistant *K. aerogenes* strains from 57 mainly small-to-medium-sized medical facilities across 20 prefectures in Japan, with 9% harboring carbapenemase genes including *bla*_IMP-6_, *bla*_IMP-1_, *bla*_NDM-1_, and *bla*_NDM-5._ Because of the small number of strains, no clear trend in the association between the STs and carbapenemase was evident. The gene *bla*_IMP-6_ was encoded on the same integron as that reported in *K. pneumoniae* in Japan, suggesting the possibility of transmission from domestic *Enterobacteriaceae*. Notably, our study uncovered a potentially novel integron with *bla*_IMP-1_ and identified similarities in the transposon structure harboring *bla*_NDM-5_ with those previously reported in Japan, suggesting that *K. aerogenes* harboring *bla*_NDM-5_ could unknowingly establish and spread domestically. These findings provide critical insights into the molecular epidemiology of carbapenemase-producing *K. aerogenes*, underscoring the need for ongoing surveillance and strategic interventions to curb the spread of these resistant pathogens.

## Figures and Tables

**Figure 1 ijms-25-04494-f001:**
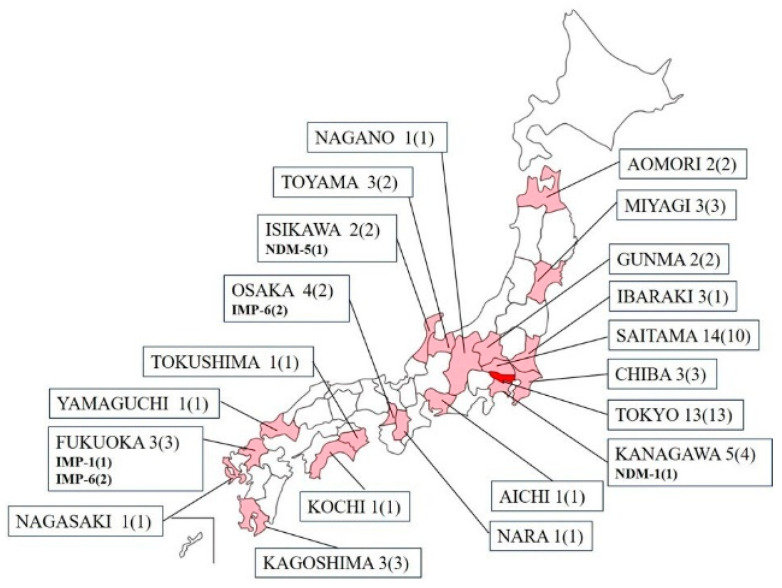
Prefectures and sample counts for collected carbapenem-resistant *Klebsiella aerogenes*. Listed are the prefectures and the number of clinical isolates (number of facilities). In prefectures where carbapenemase was detected, the type (number of isolates) is noted in bold at the bottom.

**Figure 2 ijms-25-04494-f002:**
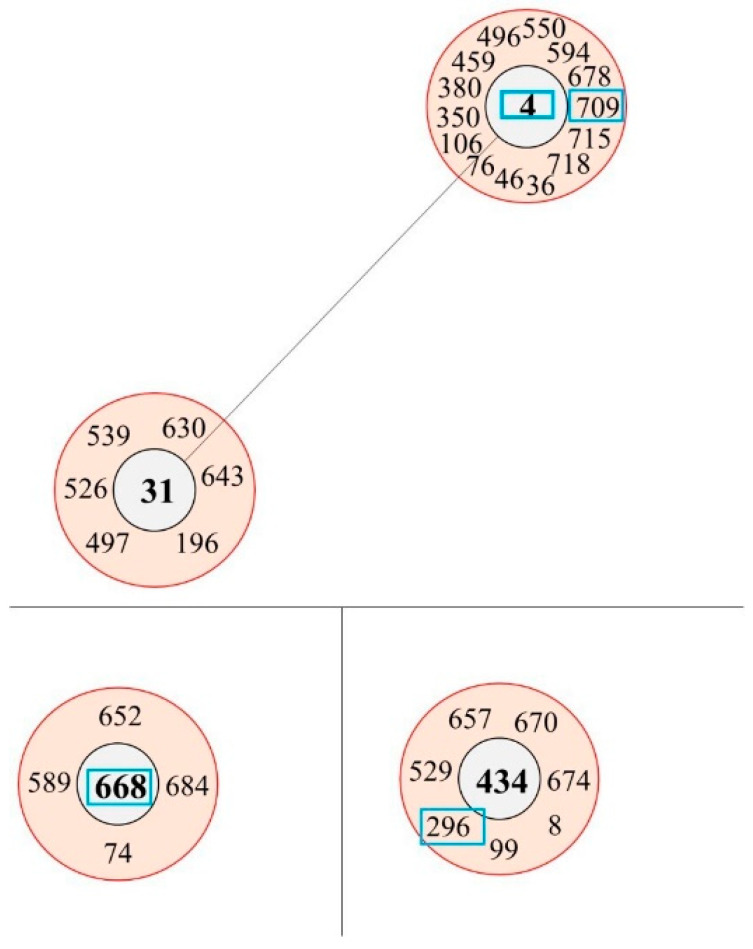
Three clonal complexes identified in this study. Three clonal complexes (CCs) were identified, including ST4, ST668, and ST296. The potential ancestral type (AT) of each CC was ST4, ST668, and ST434. CC1, centered on ST4, was the representative CC in China; ST31, although an AT, was part of the overall CC centered on ST4; ST668 and a CC containing ST296 had not been reported previously.

**Figure 3 ijms-25-04494-f003:**
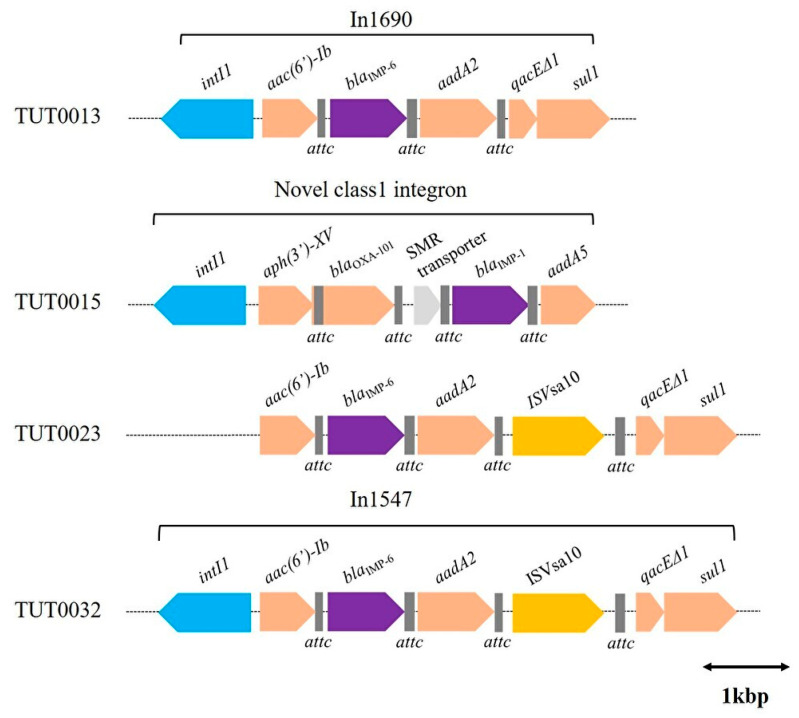
Linearized genetic environment around *bla*_IMP_. The respective integron structure, including the *bla*_IMP_, is shown. Dotted lines are nucleotide sequences; major open reading frames (ORFs) in the mobile genetic elements are indicated by blocked arrows. Coding sequences are represented by arrows. The size of the arrow indicates the length of the ORF. An index of 1 kbp is used as a reference for the length. Arrows indicate class 1 integrase (light blue), transposase or IS element (yellow), carbapenemase gene (purple), resistance genes other than carbapenemase (beige), and others (gray). Squares indicate attC.

**Figure 4 ijms-25-04494-f004:**
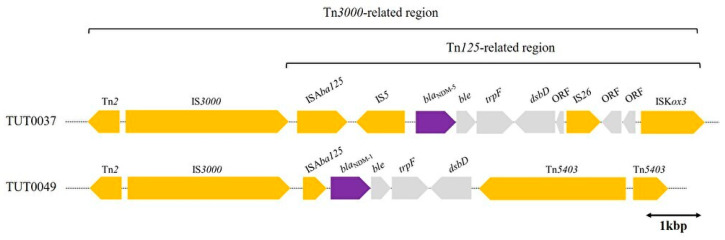
Linearized genetic environment around *bla*_NDM_. Gene structures around *bla*_NDM_ in TUT0037 and TUT0049. Both TUT0037 and TUT0049 had transposases, such as IS*3000* and ISA*ba125* upstream of *bla*_NDM_. ISA*ba125* and IS*3000* constituted transposons Tn*125* and Tn*3000*, respectively. Regions considered to be related to Tn*3000* are labeled “Tn*3000*-related regions” and regions considered to be related to Tn*125* are labeled “Tn*125*-related regions”. The orientations and colors of the block arrows are the same as in Figure 3.

**Table 1 ijms-25-04494-t001:** Basic information on six Klebsiella aerogenes strains harboring carbapenemase, along with genetic testing and antimicrobial susceptibility testing results.

Strain Name	Year of Isolation	Prefecture	Type of Facility	Age (Sex)	Sample	STs	Carbapenemase Gene Type	Replicon Types	Integron and Transposon	Antimicrobial Resistance Genes Except Carbapenemase Gene
TUT0013	2016	Fukuoka	Acute	89 (M)	Bile	189	*bla* _IMP-6_	–	In1690	*aadA2*, *aac(6′)-Ib*, *sul1, bla*_SHV-190_, *oqxA*, *tet(A)*, *bla*_CTX-M-2_
TUT0015	2016	Fukuoka	Long-term	87 (F)	Sputum	4	*bla* _IMP-1_	IncN_4_	Novel class 1 integron	*bla_OXA-101_*, *aph(3′)-XV*, *fosA*, *qnrS1*
TUT0023	2017	Osaka	Acute	67 (M)	Urine	668	*bla* _IMP-6_	–	In1547-like	*aadA2*, *aac(6′)-Ib*, *sul1, fosA7*, *dfrA14*, *bla_CTX-M-2_*
TUT0032	2018	Osaka	Acute	84 (F)	Sputum	668	*bla* _IMP-6_	–	In1547	*aadA2*, *aac(6′)-Ib*, *sul1, fosA7*, *dfrA14*, *bla_CTX-M-2_*
TUT0039	2018	Ishikawa	Acute	72 (M)	Urine	709	*bla* _NDM-5_	IncX_3_	Tn3000-like	*qnrD1*
TUT0049	2019	Kanagawa	Acute	72 (F)	Vaginal discharge	296	*bla* _NDM-1_	IncX_3_	Tn3000-like	*qnrD1*

## Data Availability

The datasets created and analyzed in the current study are not publicly available because they contain a large amount of detailed patient information. The dataset was owned by the Department of Infectious Diseases, Internal Medicine, Tohoku University Graduate School, and BML Inc., Tokyo, Japan.

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
