# Peer review of "Molecular Epidemiology of Carbapenem-Resistant Klebsiella aerogenes in Japan"

_ijms, 2024, doi:10.3390/ijms25084494_

Round 1

Reviewer 1 Report

Comments and Suggestions for Authors

The paper deals with the genetic analysis of carbapenemase-producing cedpas in a multicenter study in Japan and gives a detailed description of the phenomenon. The methodology employed is adequate and novel to achieve the objectives. 

The discussion should be improved by comparing the data with other geographical areas and with other species of enterobacteria producing the same enzymes.

Author Response

Response to the reviewer

Thank you for pointing out. The strains possessing IMP-type carbapenemase were compared with other bacteria, both domestic and foreign. Because of the complexity and redundancy of the details in the text, a supplemental file was created and a list of the cited information is included in the file. NDM-type carbapenemases were also compared to sequences with similar structures at home and abroad.

Reviewer 2 Report

Comments and Suggestions for Authors

The manuscript describes the molecular epidemiology of K. aerogenes carrying carbapenemase genes in Japan. Surveillance studies play a vital role in monitoring ARG and shed light on the global scenario, now recognized as a silent pandemic. Overall, the manuscript demonstrates commendable organization and writing. However, I have made some comments that may enhance and better align the manuscript for its final version.

Line 31: remove "facultative anaerobic bacteria" and keep "facultative bacteria." 

Line 55 - 57: this statement seems fragmented. I suggest rewriting it to:
"Epidemiological studies conducted in major Japanese hospitals support this trend [11], indicating that carbapenemase-producing K. aerogenes strains are extremely rare or non-existent".  

-  Since the authors have included a region collection map, I recommend incorporating clinical information, especially concerning the 6 isolates. Furthermore, please add the source of isolation for these 6 isolates (e.g., blood, urine, fecal swab). Additionally, clarify whether these samples represent general patient infections or surveillance swabs.

 Line 103 and 120: I recommend reordering these sections.
 Did the authors perform MLST first and then check the carbapenem genes? A well-organized workflow is essential for the presentation of the results.

 - Please refer to Table 1 in sections 2.1 and 2.3 of the text

- Perform pMLST for the IncN plasmids.

- Add virulence genes to this analysis. 

- How did the authors verify that the integron from TUT0012 was accurately assembled? Have they submitted the new integron sequence to the INTEGRALL database? If not, I strongly recommend doing it.

- Given the limitation of short reads highlighted by the authors, conducting mobilization assays is necessary to overcome this constraint and comprehensively understand the plasmid biology of these strains. I suggest performing circularization or mobilization assays to confirm the location of the antibiotic resistance genes, particularly the blaIMP-6 gene

Comments on the Quality of English Language

minor English edition

Author Response

Reviewer2

Point-by-point responses to the Reviewer’s comments

Thank you for taking the time to review our manuscript and provide insightful and helpful comments, despite your busy schedule. We sincerely appreciate it. We have made revisions to the content of the paper based on your valuable comments. We kindly request you to review it once again. Thank you in advance for your assistance.

1.Reviewer's suggestion or questions
Line 31: remove "facultative anaerobic bacteria" and keep "facultative bacteria." 
Response to the reviewer

I have corrected it as you suggested.

2.Reviewer's suggestion or questions
Line 55 - 57: this statement seems fragmented. I suggest rewriting it to:
"Epidemiological studies conducted in major Japanese hospitals support this trend [11], indicating that carbapenemase-producing K. aerogenes strains are extremely rare or non-existent".  

Response to the reviewer

I have corrected it as you suggested. I feel that it has improved the connection between the sentences.

3.Reviewer's suggestion or questions
-  Since the authors have included a region collection map, I recommend incorporating clinical information, especially concerning the 6 isolates. Furthermore, please add the source of isolation for these 6 isolates (e.g., blood, urine, fecal swab). Additionally, clarify whether these samples represent general patient infections or surveillance swabs.

Response to the reviewer

As you pointed out, we have incorporated clinical information such as patient age and gender for the six isolates. We have also incorporated the source of the isolates, which, as noted in 2.1, are all clinical isolates and were not collected specifically for surveillance purposes.

4.Reviewer's suggestion or questions

Line 103 and 120: I recommend reordering these sections.
 Did the authors perform MLST first and then check the carbapenem genes? A well-organized workflow is essential for the presentation of the results.

Response to the reviewer

Thank you for pointing this out. Initially, we thought that the resistant gene, plasmid, and mobile genetic elements items together would make a better connection in the text. After discussing with my co-authors, it seems that the interchange of 2-2 and 2-3 is not a problem, and we have made the change as you suggested.

 ï¼•ï¼ŽReviewer's suggestion or questions 

Please refer to Table 1 in sections 2.1 and 2.3 of the text

Response to the reviewer

Sorry. I don't know if I was able to adequately frame your intent, but I added references to Table 1 in Sections 2.1 and 2.3 (2.2 after revision). The reference to Table 1 was added in Figure 1, which is mainly included in Section 2.1.

6.Reviewer's suggestion or questions 

- Perform pMLST for the IncN plasmids

Response to the reviewer

Thank you for your important point, I have analyzed the TUT0013-NODE137  file containing IncN with plasmidMLST and the repN is confirmed. However, the important point is that the NODE containing carbapenemase was NODE173. In other words, we could not tell if the plasmid harboring the carbapenemase was IncN. This is a misstep that begins with a simple misunderstanding of 137 and 173, but it has a significant impact on the following text. In the following text, the sentence about IncN has been largely revised and deleted. For L155 to L160, the content was modified because IncN is not on the same contig as blaIMP-6.

7.Reviewer's suggestion or questions 

- Add virulence genes to this analysis. 

Response to the reviewer

Thank you for your interesting point, however, our research focuses on the details of carbapenemase possession and mobile genetic elements such as integrons in K. aerogenes, and it is beyond the scope of this study to perform the PCR test again to identify virulence genes.

8.Reviewer's suggestion or questions 
- How did the authors verify that the integron from TUT0012 was accurately assembled? Have they submitted the new integron sequence to the INTEGRALL database? If not, I strongly recommend doing it.

Response to the reviewer

Sorry. The TUT0012 you pointed out does not exist, so I will assume TUT0015 in my reply. the extent of integrons was confirmed by analyzing contigs containing blaIMP-1 with Integronfinder 2.0. The structure and sequence of each gene were identified with Bioinfomatic tools such as Resfinder and VRprofile2. Integron-containing contigs were checked with NCBI blast to ensure that no similar structures were registered. Is this a response to your question?

we considered registering it in Integrall. The contact person was not feeling well and gave me another contact person, the second contact person sent me an email saying a third contact person would contact me, but I never heard from them. Since then, We have been trying to contact them to register but have been unable to reach them; the Integrall situation has also not been updated since 2021.

9.Reviewer's suggestion or questions 
- Given the limitation of short reads highlighted by the authors, conducting mobilization assays is necessary to overcome this constraint and comprehensively understand the plasmid biology of these strains. I suggest performing circularization or mobilization assays to confirm the location of the antibiotic resistance genes, particularly the blaIMP-6 gene

Response to the reviewer

Thank you for your important suggestion. This study focuses on the identification of peripheral structures, including blaIMP and blaNDM, for comparison with domestic and foreign sequences. The integrons were in the same contig as the carbapenemase gene and range determination was possible. Hence, some modifications were made to the Limitation. As for the transposons, it is not clear whether short-read sequencing has completely reproduced the transposons or not. However, this study was not focused on the identification of transposons, and therefore, it is beyond the purpose of the study to attempt to reproduce transposons using long-read sequencing. Hence, despite the limitations of short-read sequencing as described in Limitation, we believe that we were able to make comparisons among bacterial species and within and outside of Japan. However, we acknowledge the significance of your suggestion in advancing our understanding of the genetic context and mobility of antibiotic resistance genes. We agree that such assays could provide valuable insights and will consider incorporating mobilization assays in future studies to better elucidate the plasmid biology and gene mobility.

Reviewer 3 Report

Comments and Suggestions for Authors

l. 32 – Enterobacterales is order, family is Enterobacteriaceae

l. 31 – 81 – justify text

l. 38 - instead „central“ use word „main“

l. 33, 41, 44 .. – unify writing in italics E. cloaceae complex

l. 61 – missing space in front of the bracket

l. 78 – there is one space more after...“In this study,“

l. 78 – I recommend to make a new paragraph strating with „In the study,“ to make the aim of this work clear

l. 84 –I suggest to shorten and specify more the heading of the chapter 2.1 to be more accurate and precise

l. 103 – „K. aerogenes“ should be in italics too

l. 184 – „linearized“ should start with capital letter

l. 204–209 - justify text and there is missing gap between previous line with the Fig. heading

l. 221 – „K. aerogenes“ should be in italics too

l. 244–266 - justify text

l. 253, l. 275, l. 288, l. 319 – missing space in front of the bracket

l. 406–422 - justify text

l. 416 – I recommend to add e.g. „The gene bla...“ in front of the gene name to start the sentence

throughout the document – check missing/remaining gaps in the text (probably l. 188, l. 196, l. 261, l. 314...)

Author Response

Point-by-point responses to the Reviewer’s comments

Thank you for taking the time to review our manuscript and provide insightful and helpful comments, despite your busy schedule. We sincerely appreciate it. We have made revisions to the content of the paper based on your valuable comments. We kindly request you to review it once again. Thank you in advance for your assistance.

Reviewer3

  1. Reviewer's suggestion or questions
  2. 32 – Enterobacterales is order, family is Enterobacteriaceae

Response to the reviewer

We chose “enterobacterales” because the CRE description has recently been changed to carbapenem-resitant Enterobacterales. However, as you pointed out, it is not a family, so we deleted “family”.

  1. Reviewer's suggestion or questions
  2. 31 – 81 – justify text

Response to the reviewer

I have corrected it as you indicated.

  1. Reviewer's suggestion or questions
  2. 38 - instead „central“ use word „main“

Response to the reviewer

I have corrected it as you indicated.

  1. Reviewer's suggestion or questions
  2. 33, 41, 44 .. – unify writing in italics E. cloaceae complex

Response to the reviewer

I have corrected it as you indicated.

  1. Reviewer's suggestion or questions
  2. 61 – missing space in front of the bracket

Response to the reviewer

I have corrected it as you indicated.

  1. Reviewer's suggestion or questions
  2. 78 – there is one space more after...“In this study,“

Response to the reviewer

I have corrected it as you indicated.

  1. Reviewer's suggestion or questions
  2. 78 – I recommend to make a new paragraph strating with „In the study,“ to make the aim of this work clear

Response to the reviewer

Thank you for pointing this out. I have corrected it as you indicated.

  1. Reviewer's suggestion or questions
  2. 84 –I suggest to shorten and specify more the heading of the chapter 2.1 to be more accurate and precise

Response to the reviewer

As you indicated, we have made the title short and concise.

  1. Reviewer's suggestion or questions
  2. 103 – „K. aerogenes“ should be in italics too

Response to the reviewer

I have corrected it as you indicated.

  1. Reviewer's suggestion or questions
  2. 184 – „linearized“ should start with capital letter

Response to the reviewer

I have corrected it as you indicated.

  1. Reviewer's suggestion or questions
  2. 204–209 - justify text and there is missing gap between previous line with the Fig. heading

Response to the reviewer

I have corrected it as you indicated.

  1. Reviewer's suggestion or questions
  2. 221 – „K. aerogenes“ should be in italics too

Response to the reviewer

I have corrected it as you indicated.

  1. Reviewer's suggestion or questions
  2. 244–266 - justify text

Response to the reviewer

  1. Reviewer's suggestion or questions
  2. 253, l. 275, l. 288, l. 319 – missing space in front of the bracket

Response to the reviewer

I have corrected it as you indicated.

  1. Reviewer's suggestion or questions
  2. 406–422 - justify text

Response to the reviewer

I have corrected it as you indicated.

  1. Reviewer's suggestion or questions
  2. 416 – I recommend to add e.g. „The gene bla...“ in front of the gene name to start the sentence

Response to the reviewer

I have corrected it as you indicated.

  1. Reviewer's suggestion or questions

throughout the document – check missing/remaining gaps in the text (probably l. 188, l. 196, l. 261, l. 314...)

Response to the reviewer

I have corrected it as you indicated.

Reviewer 4 Report

Comments and Suggestions for Authors

This study highlights the concerning rise of carbapenem-resistant Klebsiella aerogenes strains in Japan, underscoring the urgent need for vigilant surveillance and targeted interventions. The identification of novel strains and the presence of diverse carbapenemases, including NDM variants, emphasize the complexity of this issue. Understanding the molecular epidemiology and mechanisms of resistance is crucial for effective containment strategies. The findings underscore the importance of global collaboration and ongoing research to combat the spread of multidrug-resistant pathogens and safeguard public health.

However, I found the manuscript to be okay, but it was very hard to follow. Especially the results section; What question are you asking? What methods did you use to answer this question? What results did you get? What is the conclusion? It will help to improve the manuscript.

Abstracts also need to improve.

Author Response

Point-by-point responses to the Reviewer’s comments

Thank you for taking the time to review our manuscript and provide insightful and helpful comments, despite your busy schedule. We sincerely appreciate it. We have made revisions to the content of the paper based on your valuable comments. We kindly request you to review it once again. Thank you in advance for your assistance.

Reviewer4

Reviewer's suggestion or questions

This study highlights the concerning rise of carbapenem-resistant Klebsiella aerogenes strains in Japan, underscoring the urgent need for vigilant surveillance and targeted interventions. The identification of novel strains and the presence of diverse carbapenemases, including NDM variants, emphasize the complexity of this issue. Understanding the molecular epidemiology and mechanisms of resistance is crucial for effective containment strategies. The findings underscore the importance of global collaboration and ongoing research to combat the spread of multidrug-resistant pathogens and safeguard public health.

However, I found the manuscript to be okay, but it was very hard to follow. Especially the results section; What question are you asking? What methods did you use to answer this question? What results did you get? What is the conclusion? It will help to improve the manuscript.

Abstracts also need to improve.

Response to the reviewer

As you pointed out, the results were difficult to understand and follow up. Some corrections were made to the content. The second half of the introduction, which provides the basis for the results and conclusions, has been revised to make the study and its objectives more clear. The results and discussion were summarized in a supplementary file for the cited integrons to make them easier to follow and read. Some additions were made to the conclusions to take this into account.

Round 2

Reviewer 2 Report

Comments and Suggestions for Authors

The study is epidemiologically significant in monitoring carbapenem-resistant K. aerogenes in Japan. However, the data presented lacks originality, and the authors have not sufficiently addressed my suggestions.

Reviewer 4 Report

Comments and Suggestions for Authors

no further comments. The manuscript is good for publication.